# Searching for Novel Sustainability Initiatives in Amazonia

**Gabriel Medina** [1,*]**, Cassio Pereira** [2]**, Joice Ferreira** [3]**, Erika Berenguer** [4] **and Jos Barlow** [5]

1    Faculty of Agronomy and Veterinary Medicine, University of Brasilia, Brasília 70910-900, Brazil
2    Iniama, Belém 66095-105, Brazil
3    Embrapa, Belém 66095-903, Brazil
4    Ecosystems Lab, University of Oxford, Oxford OX1 2JD, UK
5    Lancaster Environment Centre, Lancaster University, Lancaster LA1 4YW, UK
*    Correspondence: gabriel.medina@unb.br

**Abstract:** Amazonia is facing growing environmental pressures and deep social injustices that prompt questions about how sustainable development may emerge. This study sought novel sustainability initiatives in the Brazilian Amazon based on interviews conducted with diverse practitioners in 2021 using a horizon-scanning approach and snowball sampling for selecting interviewees, who then described the initiative most familiar to them. The interviews resulted in 50 described initiatives and 101 similar initiatives that were listed but not described. The results reveal the emergence of a range of sustainability initiatives, which we classify into seven types of new seeds of change ranging from eco-business opportunities, territorial protection by grassroots movements, and novel coalitions promoting sustainability. However, most of these new seeds are still being established and have a limited or uncertain potential for replication, and most offer only incremental rather than transformative development. Therefore, although these initiatives provide weak yet real signals for alternative futures, they also suggest that much more needs to be done to support the needed transformation toward sustainable and equitable development.

**Keywords:** sustainable development; innovative solutions; bioeconomy; new business; horizon scanning

## 1. Introduction

The question of what constitutes "sustainable development" in Amazonia is of planetary significance, yet it remains uncertain and contested. This is because fostering sustainable development has often focused on balancing the often-competing interests of industrial agriculture and infrastructure expansion with the needs and rights of forest peoples and the conservation of forests and other Amazonian ecosystems. The present challenge could not be greater, as a period of reduced deforestation (2005–2012) has been reversed, with a return to the rapid advance of the agricultural frontier and growing pressures on forest peoples in the Brazilian Amazon [1], with increased rates of deforestation [2], conflicts with local communities, and inequality and poverty [3]. These social and environmental processes are set within a political context where the consolidation of a strong lobby in favor of large-scale agricultural operations [4] has weakened environmental governance [5,6].

Despite the alarming recent trends, many of these social and environmental problems have been occurring in the Amazon for decades. Nonetheless, some of the potential strategies for ameliorating these threats are well-known. The Amazon has been a laboratory of development initiatives for over 50 years [7], and the many established approaches used to address long-standing social and environmental issues include the demarcation of indigenous territories and extractive reserves as a means of protecting forest communities and maintaining traditional, relatively sustainable uses of forest resources [8–10]; market-based initiatives such as the soy moratorium for preventing agribusiness-related deforestation [11–13]; sustainable development projects based on payments for environmental services (PES), community forestry, etc. [14–16]; and government-promoted environmental

law-enforcement and development initiatives such as public procurements from family farmers [2]. The successes and failures of many of these approaches are well-known [15], and recent syntheses suggest that many of them will continue to play a prominent role in the coming decades [17].

While established initiatives are increasingly well documented [7], there are two reasons why novel initiatives, distinct from the established set, may be emerging. First, recent societal changes and environmental challenges in the Amazon may have created a fertile space for innovation, driven by rapid changes in social conditions, technologies, and awareness of climate crises in recent years [17,18]. Second, the breadth of approaches to sustainability itself have broadened, with recent evaluations including climate and advocacy coalitions [19,20], forest restoration [21], public–private partnerships [22], and the bioeconomy [23]. Given this context, it is imperative to evaluate whether a new generation of sustainable development initiatives is emerging, as these could potentially be some of the seeds of change promoting promising future scenarios [24].

We addressed this challenge by conducting a cross-sectoral search for recently implemented initiatives broadly (and some contentiously) relevant to sustainable development in the Brazilian Amazon, with the aim of revealing a practitioner's perspective of whether there are new seeds of change. Hence, our focus is on seeds that have been put into practice, rather than just being ideas, and forms a first step in scanning for novel initiatives that could be signals of a more sustainable future for Amazonia. Specifically, this study investigates: (1) What are the emerging sustainability initiatives identified by development practitioners, who is promoting them, and what are their main novel characteristics? and (2) What are the key features of these new initiatives in terms of their level of maturity, potential for scaling up, complementarity with existing initiatives, and transformative capacity? We use these results to discuss how these novel initiatives may complement—or clash with—more-established development initiatives, and we outline the capacities of our identified seeds to either transform Amazonia through radical change or maintain the status quo, seeking a sustainable future within the existing social, political, and economic structures.

## 2. Theoretical Framework

The seeds concept can help us to understand the different components of a better future that people want and to recognize the processes that may foster the emergence and growth of solutions that fundamentally change human–environment relationships. Bennett et al. (2016) define these seeds as existing initiatives (social, technological, economic, or social–ecological ways of thinking or doing) that represent a diversity of worldviews, values, and regions but are not currently dominant or prominent in the world [24].

Our premise is that scanning for next-generation seeds is insightful because it can provide signals for pathways towards a more sustainable future, even if those signals are currently weak [25]. Identifying seeds of change can be a valuable first step in the decision-making process by creating a comprehensive and transparent basis for subsequent assessments of evidence and effectiveness and contextualised considerations for the practical implementation of different development options [26]. Sustainable development broadly relates to combining concerns for environmental and socioeconomic issues, but there are diverse understandings of what sustainable development looks like in practice. These differences are rooted in contrasting attitudes towards change and means of change, which vary across particular areas of thought [27]. Hopwood et al.'s seminal paper proposed that pluralistic understandings of sustainable development become clearer when 'mapping' initiatives along two axes: a gradient of environmental concern (from virtually none to eco-centred) and a gradient of concern for human well-being and equality.

Concern for well-being and equality among sustainable development initiatives in Amazonia is, we expect, highly variable. Hopwood's first grouping is the status quo, the view that sustainable development can be achieved within present structures. This approach emphasizes top-down management and incremental change through existing decision-making structures and hence would conceive that sustainable development in

Amazonia does not require changes in the distribution of political power, economic resources, or land. Such incremental initiatives, even if not addressing (or attempting) transformations, may still have a positive role by engaging with more downstream points of intervention [28]. To reiterate, transformative development is associated with a change in paradigm that is preceded by significant shifts in the locus of authority over policy and experimentation with new forms of development [29].

Transformative initiatives can benefit from 'leverage points' in complex systems, where relatively small changes can lead to potentially transformative systemic changes that transform the system's rules, values, and paradigms [30]. In this sense, any transformative seeds of change are a source of social innovations that rely on alternative views of nature and social relations [31]. Scholars emphasize the role of civil society in ways in which it can 'unsettle established practices and challenge the state' or how a political culture favourable to sustainability might be nurtured [32]. Speaking to the notion that seeds require nourishment, promising social and technical innovations with the potential to change unsustainable trajectories need to be nurtured and connected to broad institutional resources and responses [33]. Transformations also imply the need to create 'change agents' who can help accelerate change even in difficult circumstances [34]. Transformational change is defined as 'shifts in power relations, discursive practices, and incentive structures that lead away from unsustainable and unjust exploitation' [35].

Following this theoretical framework, we defined an initiative as the project, practice or process mentioned by the interviewed expert and seeds as categories used by the research team to group similar initiatives. New seeds were defined as emerging initiatives that exist and that represent a diversity of worldviews, values, and regions but are not currently dominant or prominent in the world [24]. Finally, we defined 'seeds of transformational change' as initiatives that are associated with a change in the development paradigm [29].

### 3. Materials and Methods

We used a solution-scanning (or horizon-scanning) approach to make a first evaluation of the new initiatives. Solution scanning involves listing all the known options for addressing a particular problem [36], and it is the first stage of the subject-wide synthesis of evidence [25]. While a complete review of the evidence base for all available initiatives would be preferable, the scale and duration of such reviews are often impractical [26], especially where the solutions are not yet fully developed [37]. Such complete reviews are impractical for something new, as there is, almost by definition, scant or no literature on 'new seeds'.

For scanning the new initiatives in the Amazon, we used a classical snowballing approach as a means for selecting new initiatives. Specifically, interviewees were asked to describe the initiative they were more familiar with and also to mention other cases and experts who could be further contacted by the research team. Every interviewed person was asked to list the new initiatives they had heard about and then describe the one he/she knew best. The research team followed the received suggestions until the moment when new mentioned initiatives (or seeds that were equivalent to them) were already described in our database.

We used two complementary approaches for selecting practitioners and initiatives: (1) a list of key experts, in which we began with people we had personal contact with and asked about contact of other people they knew, and (2) a list of initiatives, in which we also began with people connected to the initiatives we knew and followed the received suggestions. This study is based on 52 interviews with diverse experts including forest peoples, rural smallholders, and their representatives (such as farmers' unions); NGO managers; government agents; and scholars. Interviews were conducted between July and December 2021, resulting in 50 initiatives fully described and 101 different initiatives listed but not described. All but two respondents described one initiative he/she considered to be novel.

A definition of new initiatives was provided upfront to the interviewees, who then listed the initiatives that they considered to fall under these criteria. Interviewees then selected one case to be described in detail. New initiatives were defined as novel on-going efforts towards sustainable development that were different from long-standing initiatives. We invited the interviewees to use their own definition/understanding of sustainable development.

All interviews were conducted with adults (>18 years) from across a range of sectors and geographic sub-regions in Brazilian Amazonia (Table 1). The sampling bias towards adult male respondents who work for NGOs and are based outside the Amazon or in the states of Pará and Amazonas was an outcome of the snowball approach adopted. To protect the identity of the interviewees, we de-identified participants by creating different alpha-numerical codes for each initiative and respondent. Given that the COVID-19 pandemic severely restricted travel and opportunities for in-person contact in Amazonia, interviews were conducted remotely using unrecorded video calls on computers.

**Table 1.** Profile of the 50 interviewed sustainable development practitioners who described the initiatives reported in this study.

| Variables | Profile | Number of Interviewed Experts |
|---|---|---|
| Sector | Grassroots (rural smallholders and their representatives) | 6 |
| | NGOs | 23 |
| | Government | 8 |
| | Private | 7 |
| | Academia | 6 |
| Gender | Man | 35 |
| | Woman | 15 |
| Focus area | National level (Brazil) | 15 |
| | Regional (Amazonia) | 11 |
| | Pará State | 11 |
| | Amazonas State | 11 |
| | Other Brazilian Amazonian states | 2 |

In order to characterize the initiatives, the research team followed an interview protocol structured around four main topics: (1) the initiative's main features and then the authors' evaluations of an initiative's (2) relative power/capacity to address existing threats (i.e., varying degrees of environmental and social concerns as identified by the institution(s) leading the initiative); (3) realistic opportunities for complementing other sustainable development solutions (i.e., those not identified as novel seeds and instead likely to be dominant or prominent for attempts to resolve particular social or environmental problems); and (4) potential for scaling up in terms of area covered and people potentially benefiting from it. Key features of the initiatives included who is promoting the initiative, its aim, and what made it novel from the interviewee's perspective (i.e., unconventional and innovative rather than replicating existing ideas in Amazonia in a new location). The initiative's opportunities for complementing existing solutions was assessed based on its area of influence and whether it spatially overlaps with protected areas, rural settlements, etc. Finally, each initiative's potential for scaling up was assessed with questions regarding its launch date and its growth since then (i.e., assuming that past growth is a reliable indicator of future growth potential). For each topic, a group of questions was asked following a standardized research protocol.

We grouped these initiatives after the interviews into categories based on the problems initiatives are setting out to solve, the solutions they are attempting to achieve, and the main sector involved. Single-sector grouping proved infeasible since some sectors such as NGOS are involved in so many of these activities. Since some initiatives would fit in

different seeds, we acknowledge other forms of categorization. Given the specificities of the initiatives, some categories were given subcategories.

## 4. Results

### 4.1. The New Initiatives

This section presents the new initiatives identified by interviewees as well as their main promoters and claimed novelties. Figure 1 presents the spatial distribution of the described initiatives based on the location of the headquarters of their main institutional promoter.

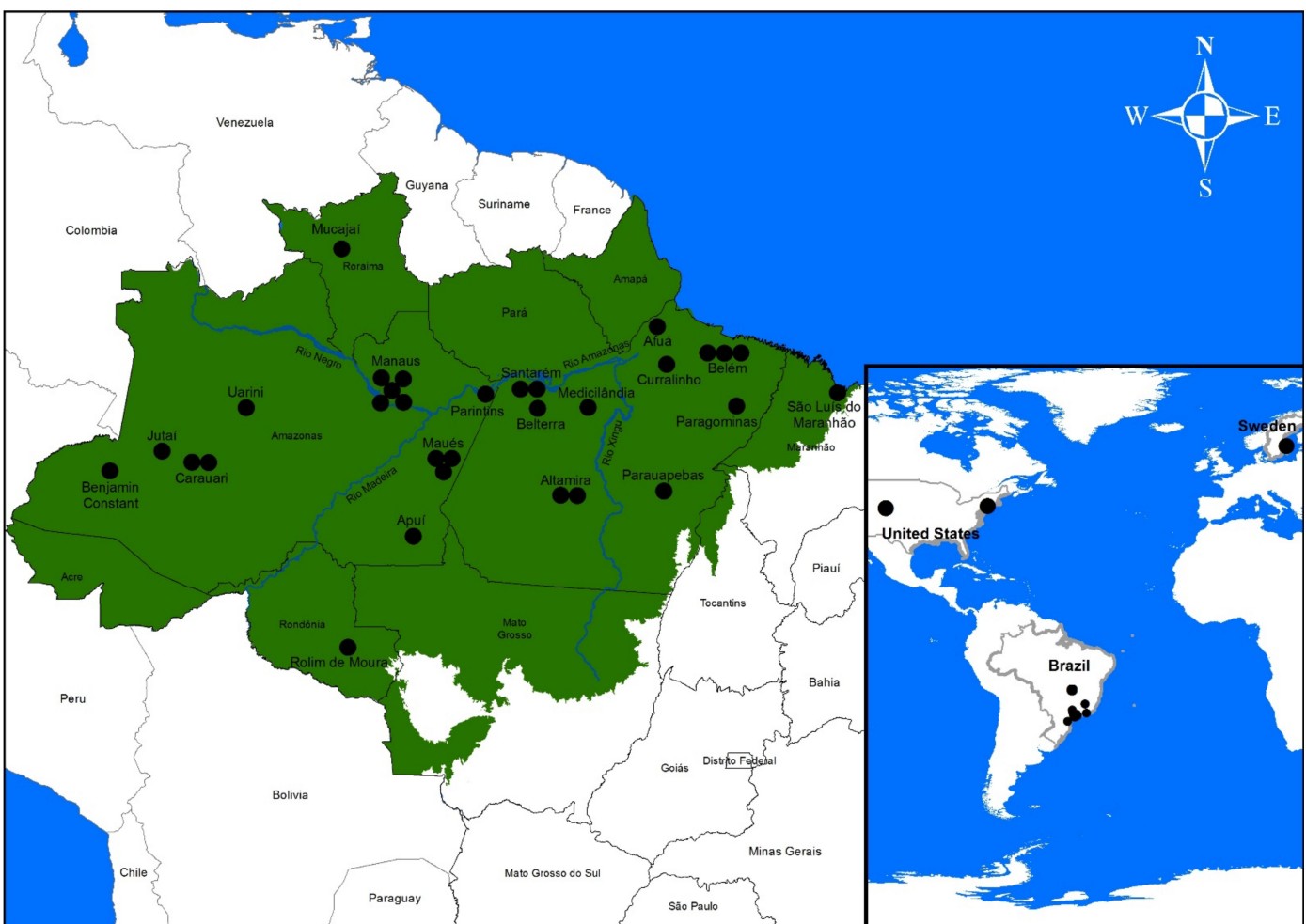

**Figure 1.** Locations of the described new sustainable development initiatives in the Brazilian Amazon.

4.1.1. What Kinds of New Initiatives Did the Interviewees Identify?

The interviews resulted in 50 described and 101 similar initiatives that were listed but not described. The start dates of these initiatives were from 2007 to 2021, with 44% starting just three years before the interviews, and 88% were within the last six years. The identified new initiatives were summarized into seven groups:

(1) **Eco-business opportunities**—Initiatives related to creating new business opportunities for products from the Amazon. These seeds encompass bioeconomy business incubators; processing high-value products for niche markets; and innovative marketing hubs. These seeds include 13 described and 22 listed initiatives.

(2) **Environmental and social accountability in agribusiness supply chains**—Initiatives related to improving traceability and farming practices in the agribusiness commodities supply chains such as for soybeans and cattle. This group of seeds encompasses transparency and improved field practices and includes 3 described and 22 listed initiatives.

(3) **Private investment in sustainable development**—Initiatives by private companies either as a private investment for profit or responding to environmental or market conditions. These seeds encompass environmental/business conditions and funds and investments and include 7 described and 22 listed initiatives.

(4) **Territorial protection by grassroots movements**—Initiatives by farmers' unions or indigenous associations. These seeds encompass territorial protection and socioeconomic development initiatives and includes 4 described and 18 listed initiatives.

(5) **Subnational governmental policies**—Initiatives by autonomous federal agencies (mainly the ones that are part of the judicial power and not dependent on the President's office or a particular ministry in Brasilia) as well as by state and municipal governments. These seeds encompass initiatives related to socioeconomic development, technological development, and environmental protection. They include six described and seven listed initiatives.

(6) **Coalitions that promote rights and environment**—Sets of different stakeholders collaborating in search of sustainable development alternatives. These seeds encompass either international or national coalitions and include four described and five listed initiatives.

(7) **Civil society activism**—Initiatives by individuals or third-sector organizations. These seeds encompass initiatives related to digital activism; forest restoration and payments for environmental services; alliances for governance; and other efforts. They include 13 described and 24 listed initiatives.

### 4.1.2. Which Institutions Were Promoting the New Initiatives?

The described new initiatives were promoted by different place-based stakeholders such as governments (n = 10); NGOs (n = 21); rural communities, individuals, and social movements (n = 10); and private companies (n = 9). Some of the initiatives' costs were covered with people's own out-of-pocket money (n = 12), but a considerable number of them relied on external support (n = 38).

Key external sponsors include international cooperative organisations (n = 16) such as USAID and the Norwegian NORAD; private companies (n = 11) such as the banks Bradesco, Itaú, and Santander, and the cosmetics company Natura; Brazilian state and federal governments (n = 5) such as the states of Amazonas and Pará and the Ministry of Agriculture, or other sources (n = 6).

### 4.1.3. What Did the Interviewees Claim as Novelty?

We used our groupings to examine novelty based on the assessments of the interviewees. Based on the initiatives' main perceived innovations (Figure 2), we summarized seven types of novelty presented in the described initiatives:

(1) **Bioeconomy as a business opportunity for sustainable development**—Interviewed sustainable development practitioners emphasized the current effort to promote bioeconomy mainly through eco-business. The 13 described and 22 listed eco-business opportunities were mainly focused on tapping the green market by exploring the bioeconomy as a business opportunity for sustainable development.

(2) **New use of technology**—New technologies and solutions such as blockchain, crowdfunding, big data, and cell phone apps were used to connect development projects and sponsors and to increase transparency in supply chains in initiatives such as Nature Invest and *Do Pasto ao Prato*. Digital activists included NGOs and unions that organized on different kinds of social media as well as in demonstration campaigns in favour of conservation, fighting deforestation, and denouncing illegalities. In some ways, these efforts can be seen as an extension of previous initiatives that occurred decades ago such as Sting's activism with the Kayapo people and traditional media campaigns but using social media. However, there are two key differences. First, these latest initiatives use social media for the large-scale mobilization of civil society as a means for demanding actions from governments, private-sector companies, etc.

Second, our interviews revealed an emergence of the influencer/activist on social media, especially by indigenous women.

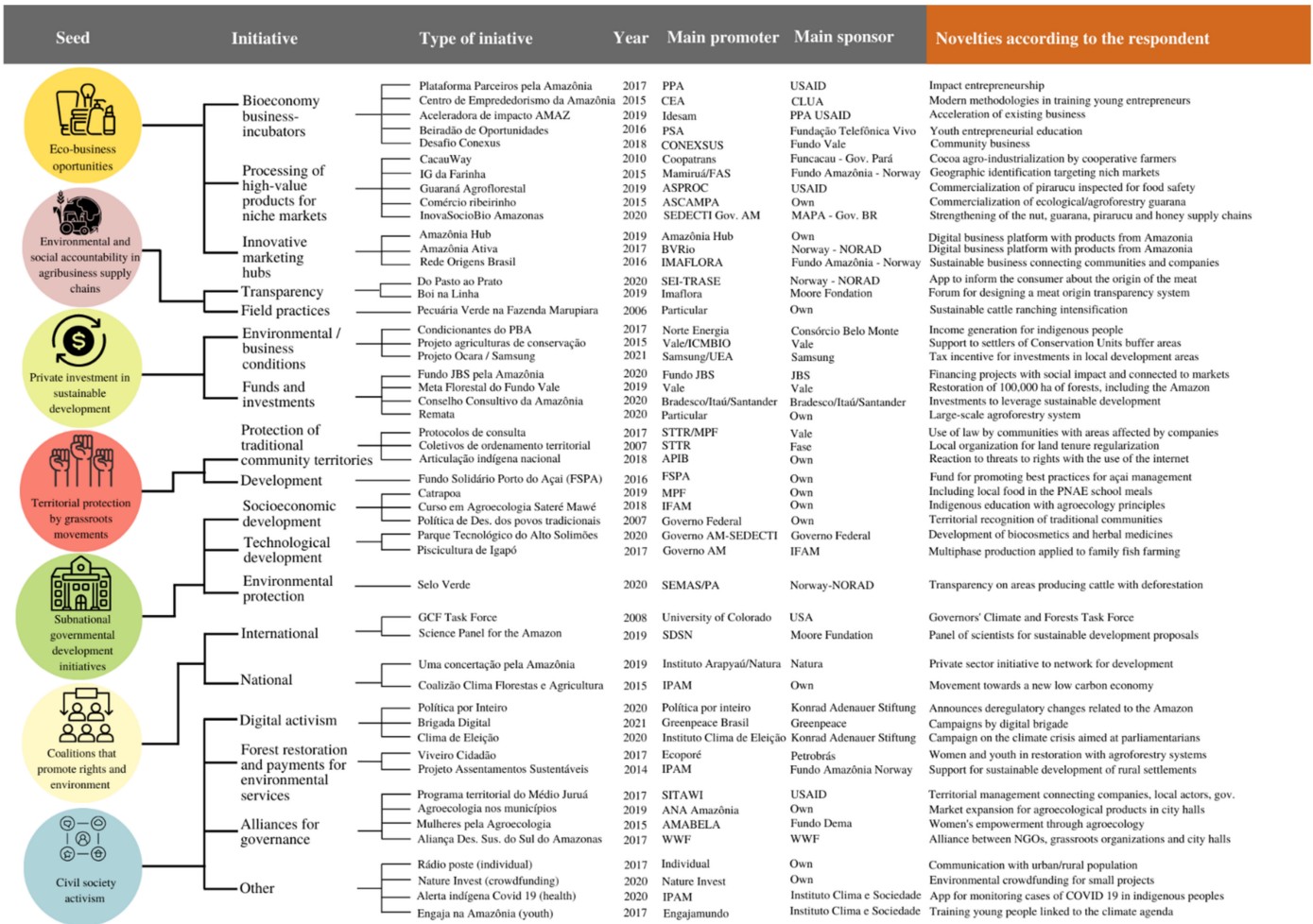

**Figure 2.** An overview of the new sustainable development initiatives identified in the Brazilian Amazon.

(3) **New connections among actors and across boundaries**—The interviews revealed a number of new coalitions that promote rights and environmental protection, including new multi-stakeholder and relatively egalitarian assemblies and decision-making forums. These contrast with former initiatives that were more limited in scope and less horizontal in terms of power balance. Much of this current mood comes from the fact that key stakeholders identified clear limits in individual actions and realized the need and the benefits of cooperation, which includes an opportunity to exert soft power and influence dynamics. International coalitions range from academic coalitions such as the Science Panel for the Amazon and political coalitions such as the GCF Task Force, initially promoted by the republican former governor of California Arnold Schwarzenegger and the Leaf Coalition recently promoted by the democratic national US government. National coalitions include private-sector initiatives to network for development such as *Uma Concertação pela Amazônia* and *Coalizão Brasil Clima Florestas e Agricultura*. The novelty is the mutual effort to cooperate in multi-stakeholder forums.

(4) **Integration of new practices into production**—Technological development initiatives include the use of biocosmetics and herbal medicines at the *Parque Científico e Tecnológico do Alto Solimões* and multiphase fish production by family farmers by the *Piscicultura de Igapó* initiative. Agribusiness sectors are also trying to respond to external demands for sustainable farming and cattle ranch intensification via initiatives that improve transparency and governance in agricultural supply chains.

(5) **New private sector actors investing in sustainable initiatives**—The private sector is promoting large-scale financial investments in support of sustainable development projects in the Amazon. These include private investments by banks, corporations, and individuals being made in forest restoration and sustainable agriculture projects such as the investments the *Conselho Consultivo da Amazônia* recommended to banks such as Bradesco, Itaú, and Santander.

(6) **New use of or creation of legislation**—Communities are promoting the local governance of territories and the use of consultation protocols (previously used mainly by indigenous peoples) as a means of territorial protection. Governmental subnational actions encompass socioeconomic development initiatives such as *Catrapoa* promoted local food in the PNAE school meals and indigenous education with agroecology principles such as the technical course in agro-ecology (*Curso técnico em agroecologia*) in the Sateré-Mawé Indigenous reserve

(7) **Emergence of grassroots actions without significant or permanent external support**—Development initiatives include funds such as the Solidarity fund of the Acai port (*Fundo Solidário Porto do Açaí*) established by local communities for financing better farming practices and market access.

Note 1: Even though some initiatives would fit different solutions, we mentioned each initiative only once in a specific category.

Note 2: Similar mentioned but not described initiatives are (101 in total): **(1) Eco-business opportunities (22 initiatives):** 1.1 Bioeconomy business-incubators: Sitawi; Amazônia UP; Ecocentro; Rainforest Social Business School; Certi Amazônia. 1.2 Processing of high-value products for niche markets: Cacau orgânico—CEPOTX; Guaraná—Sateré-Mawé; Mahá Biocosméticos; Chocolates De Mendes; Mel—Peabiru; Amazônia 4.0; Café Apui; AmazonMel; Projeto Castanha da RESEX do Rio Unini; Inatú Amazônia. 1.3 Innovative marketing hubs: Flor de Jambu; Jirau da Amazônia; Polo BioAmazonas; Observatório da Castanha da Amazônia (OCA); Amazônia 100%; Design & Madeira Sustentável; Manioca. **(2) Environmental and social accountability in agribusiness supply chains (13 initiatives):** 2.1 Transparency: Soja na Linha; Conecta; Plataforma Nice planet—SMGeo; Plataforma da JBS—rastreabilidade do gado; Sirflor (Pará)—regularização de propriedades; Plataforma de adequação ambiental do Imac (MT); Fornecedores Indiretos na Pecuária (Amigos da Terra); PrevisIA; Trase; Lucida. 2.2 Field practices: Pecuária Verde; JBS Net Zero; Pecuariando. **(3) Private investment in sustainable development (12 initiatives):** 3.1 Environmental/business conditions: Programa Prioritário de Bioeconomia; Waimiri Atroari e Parakanã—Eletronorte; Borracha—Michelin. 3.2 Funds and investments: Centro de Orquestração de Inovações (COI)—WTT; The Good Food Institute; KPTL—Fundo de Floresta e Clima; Emerge Amazônia; Impact Hub; Fundo para o Desenvolvimento Sustentável e a Bioeconomia da Amazônia; Althelia biodiversity fund; Mov investimentos; Parcerias corporativas—Café Suruí com contrato com a Três Corações. **(4) Territorial protection by grassroots movements (18 initiatives):** 4.1 Territorial protection: Campanha Não Abra Mão da Sua Terra; Protocolos de consulta sobre Asfaltamento da BR 319; Levante popular; Observatório de Segurança e Soberania Alimentar; Terra sem Males (RO). 4.2 Development: Fundo Dema; ACOSPER/UNICAFES; Projeto CAR Participativo/STTR Santarém; Programa Jurisdicional de REDD+; Finapop—MST; Flores do Campo (Grupo de mulheres de Mojuí dos Campos); Grupo de Jovens do CNS; Casa Familiar Rural de Boa Vista dos Ramos; Associações de mulheres em defesa da agricultura familiar do Rio Canaticu (Marajó); Banco Comunitário do Rio Canaticu (Moeda Social Yaça); Canindé (RO)—mobilização indígena; Comitê Chico Mendes; Surara—Coletivo de mulheres indígenas do Tapajós. **(5) Subnational government initiatives (7 initiatives):** 5.1 Socioeconomic development; Fundo de Apoio a Cacauicultura do Estado do Pará—FUNCACAU; GTI Saúde Indígena. 5.2 Technological development: Instituto Mamirauá; Rações alternativas para piscicultura—IFAM; Manejo florestal—IFT junto à Verde para Sempre; 5.3 Environmental protection: Força Tarefa Amazônia—MPF; Amazônia Protege—MPF. **(6) Coalitions that promote rights and environment (5 initiatives):** 6.1 International: The Lowering Emissions by Accelerating

Forest finance (LEAF) Coalition. 6.2 National: Consórcio de Governadores da Amazônia Legal; Amazônia 2030; Grupo Carta de Belém; Fórum Nacional Permanente em Defesa da Amazônia. **(7) Civil society activism (24 initiatives):** 7.1 Digital activism: Portal Proteja;Rede de Podcasts do Xingu; SOS Amazônia; Engajamento de influenciadores—Alok; Uma gota no oceano; Motosserra de Ouro para Arthur Lira; Floresta sem cortes; Negritar—Coletivo de audiovisual; Coletivo Jovem de Meio Ambiente Pará; 342 Amazônia; Amazônia Alerta; 7.2 Forest restoration and payments for environmental services: Aliança para Restauração da Amazônia; Projeto Bacia do Rio Putumayo-Içá; Acelerador de Agroflorestas e Restauração. 7.3 Alliances for governance: Rede Maniva de Agroecologia; Coletivo do Pirarucu; Legado Integrado da Região Amazônica—LIRA; Projeto Paisagens Sustentáveis da Amazônia. 7.4 Other: Ciência Cidadã para Amazônia; App Castanhadora; Brigada de Incêndio; Instituto Mapinguari (AP); AmIT—Amazon Institute of Technology.

### 4.2. *Features of the New Initiatives*

The new initiatives varied in terms of their level of maturity, possibility of complementing existing solutions, potential for scaling up and for transformation. Considering the initiatives' *maturity level*, out of the 50 described, 25 initiatives were reported by the interviewees as still embryonic, 21 as mature, and 4 as ended. Embryonic initiatives still in process of being implemented have not yet had the chance to address existing environmental and/or social challenges and cannot yet be considered viable or long-lasting.

Considering their capacity to *complement* existing development initiatives, 44 initiatives were deemed as building on and complementing existing environmental conservation efforts, while 6 did not include conservation measures and focused only on socioeconomic development, in the sense of concern for improving well-being but not necessarily reducing social inequities. Out of the 44 initiatives we considered to be complementary, 23 were complementary to various environmental conservation efforts, while 21 complemented specific sustainable development initiatives such as the protection of indigenous territories. Out of the 50 described initiatives, 43 claimed to support socioeconomic development by promoting better farming practices (with the perceived 'improvement' reflecting the logic of a particular initiative), sources of income, or organizational skills, while 7 did not include support for socioeconomic development and focused only on conservation. To a large extent, the identified initiatives complemented previous efforts and helped their maintenance/consolidation, such as by supporting the preservation of indigenous lands and conservation units and the development of rural settlements.

Regarding their potential to *replicate and scale up*, 28 initiatives were reported as having a replication potential limited to specific contexts, while 22 were deemed to be replicable across the whole of Amazonia. Lower-cost initiatives based on local efforts showed greater potential for replication in practice. However, initiatives such as accelerators (with greater investments) can also be promoted through public policies.

Regarding the initiative's *transformative capacity* as a whole (i.e., our analysis of whether an initiative sought radical change or instead, reform or incremental change and maintenance of the status quo), the interviewed practitioners considered that 17 presented small contributions (incremental tweaks), 29 would deliver significant contributions but without structural change (reformist adaptations), and 4 promoted structural change (radical transformations). Despite this variation, all were considered by interviewees to have some potential value for sustainable development in the Amazon. Interviewees were not asked to assess the capacity of their initiatives to deliver so-called win-wins (fulfilling socioeconomic and environmental objectives); these are hard to achieve in practice, and it is plausible that a new initiative could, for example, be effective in reducing forest degradation but simultaneously ignore or worsen social inequities.

## 5. Discussion

The large number of initiatives reported as potential new seeds of change reveal the emergence of diverse, often contradictory, pathways towards sustainable development

in the Amazon. Indeed, these initiatives are moving towards different development end-goals with respect to the weight of concern for the environment and improving human well-being and equity. Our study assessed some of these promising efforts to conceive and promote eco-business models, accountability in agribusiness supply chains, private sustainable development, grassroots rights, governmental policies, coalitions, and civil society projects. Taken together, these seeds of change connect very powerful stakeholders such as multinational companies, state governors, and large NGOs that often can count on important investments made by both domestic and international sponsors.

A key positive aspect of these new initiatives is that they often complement or build upon well-described solutions such as extractive reserves [8]. Some of these new solutions were already mentioned in other studies such as climate and advocacy coalitions [20], forest restoration [21], private investments [22], and the bioeconomy [23]. This study has attempted to systematically map out stakeholders' own perspectives on these new potential solutions, adding to previous efforts listing established initiatives in the Amazon [7].

## 5.1. Seeds' Capacity to Address Existing Environmental and Social Challenges

The new seeds are developing at a time in which the Brazilian Amazon is again experiencing growing rates of deforestation [2], growing number of conflicts with local communities [1], and increased inequality and poverty. Both classical and new sustainable development initiatives are overwhelmed by the current context.

Our assessment of the new initiatives' features reveals some key limitations regarding their ability to address existing challenges. Most initiatives are not yet completely established and have limited potential for scaling up. In this sense, we agree with previous studies' assessment that promising initiatives by themselves, despite their success in transforming local spaces, are often insufficient for advancing sustainable development at broader societal scales because the required political and environmental changes are often beyond their reach [7].

Most of these new initiatives were perceived as delivering incremental change, without upsetting the status quo of the distribution of resources and decision-making powers and falling short of offering a realistic prospect for transformative, radical change. Such incremental changes reflects literature stating that most initiatives do not address (or attempt) transformations but instead go for more downstream points of intervention [38]. In contrast, transformative initiatives tend to be context-specific and depend on 'leverage points' where relatively small changes can lead to potentially transformative systemic changes [30]. Brockhaus makes a strong case that achieving sustainable development in Amazonia and elsewhere requires confronting entrenched patterns of inequality and power relations [35]. Development success therefore requires addressing the power imbalances between different kinds of state and non-state actors. Ignoring these kinds of insights from political ecology [39] creates the conditions whereby certain kinds of seeds of sustainable development may actually reinforce and reproduce inequalities. These outcomes are difficult to predict, and incremental new initiatives could still have a positive role in supporting established initiatives.

## 5.2. Weak but Real Signals for Promising Future Scenarios

Scanning next-generation seeds is a first step towards decision-making processes that create a comprehensive and transparent basis for subsequent assessments of evidence [26]. Some of the new initiatives we have described can be seen as seeds of change that provide some weak yet rea, signals for promising future scenarios that potentially change human–environment relationships [24]. The history of the Amazon is rich in place-based solutions such as extractive reserves [9,10] that were tried out in a specific region and under specific circumstances and then expanded. The diverse approaches create an 'ecosystem' of new initiatives, increasing the chances that one or more will mature into longer-lasting and far-reaching sustainable development initiatives.

The fact that some seed types are dominant now does not necessarily reflect their capacity to bring about sustainable development but may be for circumstantial reasons. This may be the case for the growing number of efforts for forest restoration given to the UN Decade on Ecosystem Restoration as well as the 'fever' for eco-business solutions promoted by powerful stakeholders such as USAID and private companies. In the 2000s, Amazonia experienced great donor-driven support for forest management, while in the 2010s, the focus was on payments for environmental services [14]. Our assessment suggests that the bioeconomy and forest restoration driven by carbon capture could become the new donor-driven conceptual solutions for the Amazon, while bottom-up civilian science initiatives were also prevalent (Figure 2).

*5.3. Implications for Practitioners and Policy Makers*

This study reveals that most of the initiatives deemed promising and innovative by the interviewed practitioners are exogenous development concepts such as the bioeconomy that are promoted by external stakeholders such as NGOs and governmental agencies and are often sponsored by external donors. At the same time, the list of identified seeds includes grassroots initiatives mainly focused on territorial protection. Other studies highlight that endogenous, locally developed farming and governance systems should also form the basis for sustainable development in the Amazon [10]. These findings highlight the importance of considering top-down vs. bottom-up management approaches and bringing scholarly and policy attention back to local populations and their resource-use systems.

These lessons are fundamental for practitioners, policy makers, and donor agencies that should reconsider their focus on externally driven top-down concepts to provide increased support for helping communities to improve their existing systems, which are still being developed and can benefit from external support for improvements in technological, economic, environmental, and social aspects [40]. Endogenous approaches to development grounded in local practices and needs can become a viable option for sustainable development in the Amazon. For such approaches, attention should be given to farmer-led technological innovation, local governance, and the recognition of marginalized local knowledge.

*5.4. Limitations and Further Research*

The methodology used (the snowball sampling approach and the interview method) focused on the practitioners' perspectives in a qualitative evaluation of initiatives related to sustainability development and was neither comprehensive nor representative, meaning numbers should not be evaluated quantitatively. Although we did include quantitative questions on the area covered by the initiatives as well as on the number of beneficiaries, the high diversity of approaches towards sustainable development meant it was not possible to use the data collected to assess the capacity of the different initiatives to address existing threats. Some initiatives focus on raising awareness on specific issues and target different stakeholders in different areas, making it hard to identify their actual impact on the ground. Even initiatives focusing on specific areas and targeting specific communities found it difficult to measure their impacts since their actions are often part of other initiatives and dynamics.

## 6. Conclusions

The results of this study reveal the existence of new initiatives showing proof of the concept of seeds of changes that, in many cases, are complementary to existing well-known initiatives such as conservation areas, environmental law enforcement, and governments efforts to promote local development. Although these initiatives were new, their capacity to address existing threats is likely to be highly variable; most of the initiatives are difficult to evaluate as they were, by their nascent nature, not yet mature. Yet the range of seeds suggests there is a growing ecosystem of alternatives for sustainable development in the Amazon that could contribute to future solutions. However, the new initiatives are only

one part of the long-term continued efforts for sustainable development in the Amazon, and they must occur alongside other actions such as ensuring the recognition of forest peoples' access to land and resources, maintaining all of Amazonia's biodiversity, avoiding dangerous tipping points that could alter the nature of the system itself, and providing more attention to and support for endogenous, locally based development initiatives led by Amazonian rural communities and family farmers.

**Author Contributions:** G.M.—Methodology, investigation—interviews, data curation, formal analysis, writing—original draft. C.P.—Methodology, investigation—interviews, section on the origin of the initiatives, conclusion. J.F.—Conceptualization, methodology—scanning the new initiatives. E.B.—Conceptualization, methodology—scanning the new initiatives. J.B.—Conceptualization, funding acquisition, methodology, project administration, writing—original draft, writing—review & editing, classification of the initiatives. All authors have read and agreed to the published version of the manuscript.

**Funding:** This research was funded by UKRI's Global Challenges Research Fund.

**Informed Consent Statement:** Informed consent was obtained from all subjects involved in the study.

**Data Availability Statement:** Not applicable.

**Acknowledgments:** We would like to thank all the development practitioners who collaborated on this study.

**Conflicts of Interest:** The authors declare no conflict of interest.

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
