# Peer review of "Searching for Novel Sustainability Initiatives in Amazonia"

_sustainability, doi:10.3390/su141610299_

Round 1

Reviewer 1 Report

The introduction Ii too long.the objectives are not clear enough. Methods lacking statistical analysis.

Results section should have flow and presentation similar to that of methods

 The discussion section is rather weak. The conclusion can be better written

Author Response

Dear Reviewers,

We thank you for providing us with insightful recommendations for improving the manuscript, which we did our best to address in all cases as detailed below. Since major revisions were requested in some cases, we kept track changes in the manuscript so that reviewers can easily check how their recommendations were addressed.

Reviewer 1

The introduction Ii too long. the objectives are not clear enough. Methods lacking statistical analysis.

Results section should have flow and presentation similar to that of methods

 The discussion section is rather weak. The conclusion can be better written

We divided the introduction section into two different sections by including a section named theoretical framework.

A new section was added at the bottom of the discussion section for addressing your suggestion.

Reviewer 2 Report

Additional foci on the implications to practitioners in Amazonia is needed to better inform how the scholarship of the study affects practice. Examples of how to do that can be found here https://doi.org/10.3390/su131810295.

How do the authors recommend this information be communicated to policy makers to effect change? Should extension services be an actor in this process? If so, how? What about Embrapa? How could their collaboration/synergy be better leveraged to make a more positive impact? These are just suggestions but the policy makers and those leading change on the grass roots level need addressing here in order for the scholarship to be more "sustainable" and impactful. 

Recommendations for future researchers, explicitly, should be included for the next generation of 21st century scholars and those they mentor. Other research designs? What should the next research steps be and how to implemented those? 

Author Response

Dear Reviewers,

We thank you for providing us with insightful recommendations for improving the manuscript, which we did our best to address in all cases as detailed below. Since major revisions were requested in some cases, we kept track changes in the manuscript so that reviewers can easily check how their recommendations were addressed.

Reviewer 1

The introduction Ii too long. the objectives are not clear enough. Methods lacking statistical analysis.

Results section should have flow and presentation similar to that of methods

 The discussion section is rather weak. The conclusion can be better written

We divided the introduction section into two different sections by including a section named theoretical framework.

A new section was added at the bottom of the discussion section for addressing your suggestion.

Reviewer 2

Additional foci on the implications to practitioners in Amazonia is needed to better inform how the scholarship of the study affects practice. Examples of how to do that can be found here https://doi.org/10.3390/su131810295.

How do the authors recommend this information be communicated to policy makers to effect change? Should extension services be an actor in this process? If so, how? What about Embrapa? How could their collaboration/synergy be better leveraged to make a more positive impact? These are just suggestions but the policy makers and those leading change on the grass roots level need addressing here in order for the scholarship to be more "sustainable" and impactful.

Recommendations for future researchers, explicitly, should be included for the next generation of 21st century scholars and those they mentor. Other research designs? What should the next research steps be and how to implemented those?

A new section was added at the bottom of the discussion section for addressing your suggestion, in section:

5.4. Implications to practitioners and policy makers

Reviewer 3 Report

Dear Authors,

This is a well written study on bioeconomy cornerstones in Amazonia, as regarding innovative cases or exploited replication opportunities for already proven success stories. 

I suggest to include in your manuscript the limitations (if any) or bottlenecks during the study, together with further actions forseen in the future on this interesting topic, based on the collected data. 

To improve the article quality, the authors are recommended to stress out that the methodology used (the interview method) is qualitative evaluation of initiatives related to sustainability development in the area.  This is only to increase awareness that it is not quantitative study and statistical analysis has nothing to do with this kind of analysis. However, performing a quantitative study could be sugestested  in the limitations and further research section.    For helping out a bit, here are two references employing qualitative studies:   Sustainability | Free Full-Text | Models Underlying the Success Development of Family Farms in Romania (mdpi.com)

Sustainability | Free Full-Text | Urban Informal Food Traders: A Rapid Qualitative Study of COVID-19 Lockdown Measures in South Africa (mdpi.com)

  Regarding the bottlenecks, this would useful for other researchers that want to employ qualitative studies. Pointing out problems encountered and solutions found would help to further apply this research method.

Best regards,

Author Response

Dear Reviewers,

We thank you for providing us with insightful recommendations for improving the manuscript, which we did our best to address in all cases as detailed below. Since major revisions were requested in some cases, we kept track changes in the manuscript so that reviewers can easily check how their recommendations were addressed.

Reviewer 1

The introduction Ii too long. the objectives are not clear enough. Methods lacking statistical analysis.

Results section should have flow and presentation similar to that of methods

 The discussion section is rather weak. The conclusion can be better written

We divided the introduction section into two different sections by including a section named theoretical framework.

A new section was added at the bottom of the discussion section for addressing your suggestion.

Reviewer 2

Additional foci on the implications to practitioners in Amazonia is needed to better inform how the scholarship of the study affects practice. Examples of how to do that can be found here https://doi.org/10.3390/su131810295.

How do the authors recommend this information be communicated to policy makers to effect change? Should extension services be an actor in this process? If so, how? What about Embrapa? How could their collaboration/synergy be better leveraged to make a more positive impact? These are just suggestions but the policy makers and those leading change on the grass roots level need addressing here in order for the scholarship to be more "sustainable" and impactful.

Recommendations for future researchers, explicitly, should be included for the next generation of 21st century scholars and those they mentor. Other research designs? What should the next research steps be and how to implemented those?

A new section was added at the bottom of the discussion section for addressing your suggestion, in section:

5.4. Implications to practitioners and policy makers

Reviewer 3

This is a well written study on bioeconomy cornerstones in Amazonia, as regarding innovative cases or exploited replication opportunities for already proven success stories.

I suggest to include in your manuscript the limitations (if any) or bottlenecks during the study, together with further actions forseen in the future on this interesting topic, based on the collected data.

To improve the article quality, the authors are recommended to stress out that the methodology used (the interview method) is qualitative evaluation of initiatives related to sustainability development in the area.  This is only to increase awareness that it is not quantitative study and statistical analysis has nothing to do with this kind of analysis. However, performing a quantitative study could be sugestested  in the limitations and further research section.    For helping out a bit, here are two references employing qualitative studies:   Sustainability | Free Full-Text | Models Underlying the Success Development of Family Farms in Romania (mdpi.com)

Sustainability | Free Full-Text | Urban Informal Food Traders: A Rapid Qualitative Study of COVID-19 Lockdown Measures in South Africa (mdpi.com)

  Regarding the bottlenecks, this would useful for other researchers that want to employ qualitative studies. Pointing out problems encountered and solutions found would help to further apply this research method.

Addressed in section

5.5. Limitations and further research

Reviewer 4 Report

This paper provides various new seeds about sustainable development, which has very important guiding value for further research, but minor but careful revision of this paper is still needed.

(1) There are many formatting problems in this paper, which make it not so reader-friendly. For example, in Line176-183, the serial number may be preferably put in parentheses. A large number of numbers are used in this paper, and they should be displayed more clearly to make the full text more readable.

(2) The citation format needs to be standardized and unified. For example, in Line 37, [5],[6] should be [5,6]. Line 45-52, citation should be numbered rather than author-date. Line 59-61, citation should be changed to [17-22] at the end of the sentence.

(3) Table 2 is too long, maybe it can be changed into a horizontal version.

Author Response

Dear Reviewers,

We thank you for providing us with insightful recommendations for improving the manuscript, which we did our best to address in all cases as detailed below. Since major revisions were requested in some cases, we kept track changes in the manuscript so that reviewers can easily check how their recommendations were addressed.

Reviewer 1

The introduction Ii too long. the objectives are not clear enough. Methods lacking statistical analysis.

Results section should have flow and presentation similar to that of methods

 The discussion section is rather weak. The conclusion can be better written

We divided the introduction section into two different sections by including a section named theoretical framework.

A new section was added at the bottom of the discussion section for addressing your suggestion.

Reviewer 2

Additional foci on the implications to practitioners in Amazonia is needed to better inform how the scholarship of the study affects practice. Examples of how to do that can be found here https://doi.org/10.3390/su131810295.

How do the authors recommend this information be communicated to policy makers to effect change? Should extension services be an actor in this process? If so, how? What about Embrapa? How could their collaboration/synergy be better leveraged to make a more positive impact? These are just suggestions but the policy makers and those leading change on the grass roots level need addressing here in order for the scholarship to be more "sustainable" and impactful.

Recommendations for future researchers, explicitly, should be included for the next generation of 21st century scholars and those they mentor. Other research designs? What should the next research steps be and how to implemented those?

A new section was added at the bottom of the discussion section for addressing your suggestion, in section:

5.4. Implications to practitioners and policy makers

Reviewer 3

This is a well written study on bioeconomy cornerstones in Amazonia, as regarding innovative cases or exploited replication opportunities for already proven success stories.

I suggest to include in your manuscript the limitations (if any) or bottlenecks during the study, together with further actions forseen in the future on this interesting topic, based on the collected data.

To improve the article quality, the authors are recommended to stress out that the methodology used (the interview method) is qualitative evaluation of initiatives related to sustainability development in the area.  This is only to increase awareness that it is not quantitative study and statistical analysis has nothing to do with this kind of analysis. However, performing a quantitative study could be sugestested  in the limitations and further research section.    For helping out a bit, here are two references employing qualitative studies:   Sustainability | Free Full-Text | Models Underlying the Success Development of Family Farms in Romania (mdpi.com)

Sustainability | Free Full-Text | Urban Informal Food Traders: A Rapid Qualitative Study of COVID-19 Lockdown Measures in South Africa (mdpi.com)

  Regarding the bottlenecks, this would useful for other researchers that want to employ qualitative studies. Pointing out problems encountered and solutions found would help to further apply this research method.

Addressed in section

5.5. Limitations and further research

Reviewer 4

This paper provides various new seeds about sustainable development, which has very important guiding value for further research, but minor but careful revision of this paper is still needed.(1) There are many formatting problems in this paper, which make it not so reader-friendly. For example, in Line176-183, the serial number may be preferably put in parentheses. A large number of numbers are used in this paper, and they should be displayed more clearly to make the full text more readable. (2) The citation format needs to be standardized and unified. For example, in Line 37, [5],[6] should be [5,6]. Line 45-52, citation should be numbered rather than author-date. Line 59-61, citation should be changed to [17-22] at the end of the sentence. (3) Table 2 is too long, maybe it can be changed into a horizontal version.

Fixed for all the cases. The table was replaced by figures in order address your recommendation

Round 2

Reviewer 1 Report

the manuscript is improved